# Identifying Grassland Distribution in a Mountainous Region in Southwest China Using Multi-Source Remote Sensing Images

**Yixin Yuan** [1,2], **Qingke Wen** [1,*], **Xiaoli Zhao** [1], **Shuo Liu** [1], **Kunpeng Zhu** [1,2] and **Bo Hu** [1,2]

1. National Engineering Research Center for Geomatics (NCG), Aerospace Information Research Institute, Chinese Academy of Sciences, Beijing 100101, China; yuanyixin19@mails.ucas.ac.cn (Y.Y.); zhaoxl@radi.ac.cn (X.Z.); liushuo@aircas.ac.cn (S.L.); zhukunpeng211@mails.ucas.ac.cn (K.Z.); hubo21@mails.ucas.ac.cn (B.H.)
2. University of Chinese Academy of Sciences, Beijing 100049, China
* Correspondence: wenqk@aircas.ac.cn

**Abstract:** Southwest China has abundant grassland resources, but they are mainly scattered across fragmented mountainous terrain with frequently cloudy and rainy weather, making their accurate identification by remote sensing challenging. Therefore, the goal of this study was to generate prefecture-level city-scale mountainous grassland distribution data to support the development of sustainable grassland husbandry. Here, we proposed a sample selection method and comprehensively utilized multi-source data to obtain the quasi-10 m southwest grassland distribution data. The sample selection method was to first determine the sample selection range based on multi-source land use/cover database, and then to randomly select the samples under the constraint of secondary land use types, multiple factors of terrain and pure pixels. This method can deal with the difficulty in identifying the fragmented grassland distribution caused by steep mountains and hills. In addition, a multispectral time series dataset was constructed based on the fusion of Landsat 8 OLI and Sentinel-2A/B data due to cloudy and rainy weather and was used as one of the input features along with synthetic aperture radar Sentinel-1 time series data and the terrain multi-factor data. Finally, a remote sensing method to accurately identify grassland distribution in southwest China was constructed based on the Google Earth Engine (GEE) platform. Taking Zhaotong City, a prefecture-level city in Yunnan Province, as an example, a thematic map of grassland distribution with an overall accuracy of 88.21% was obtained using the above method. This map has been used by the local government of Zhaotong City in their planning of the development of sustainable grassland husbandry.

**Keywords:** southwest grassland; Google Earth Engine; time series; multi-source remote sensing identification; random forest

## 1. Introduction

Natural grassland and permanent artificial grassland (referred to as grassland) represent the largest proportion of China's terrestrial ecosystems, accounting for approximately 40% of the country's land area [1]. The preservation of grassland is important to the national goal of constructing an ecological civilization. The southern grassland is mainly distributed in the hills and mountains of 14 provinces (autonomous regions), including Yunnan, Guizhou, and Sichuan, accounting for approximately 15% of China's grassland area [2]. However, due to the presence of many mountains and hills in southwest China, grassland distribution is fragmented, which makes monitoring by field surveys or remote sensing difficult. Additionally, since the middle of the 20th century, the core of China's animal husbandry has mainly been located in the northern grassland area, with less attention given to the southwest grassland area. There have been few studies aimed at identifying the southwest grassland distribution, and the precision of the resulting data is insufficient [3]. At the same time, due to the underdeveloped information and production technology in the grassland areas of southwest China, local grazing models are inadequate, and thus the

economic and ecological benefits are still poor. Additionally, with the increasing demand for meat, eggs, and milk, and the implementation of rural vitalization strategies, there is an urgent need to promote the development of sustainable grassland husbandry in southwest China. Accurately identifying grassland distribution in southwest China can provide basic data for the development of sustainable grassland husbandry. Therefore, research on remote sensing identification methods for mountainous grassland distribution in southwest China is of great significance to environmental management and economic development at both the regional and national scale.

In addition to the difficulty in identifying the fragmented grassland distribution caused by the complex terrain, the lack of available remote sensing data due to cloudy and rainy weather in southwest China is also a challenge [4]. The existing southwest grassland distribution data mainly come from national or global land cover datasets, which leads to the problem of the inaccurate identification of grassland boundaries at the prefecture level, and there is a lack of targeted research on the remote sensing identification of southwest grassland distribution [5]. Therefore, it is necessary to overcome the above difficulties through constructing a remote sensing method to identify fragmented grassland distribution accurately.

Generally, the accuracy of image classification depends largely on sample selection and input feature selection [6,7].

The spatial uniformity of sample selection makes it representative and reduces the problem of excessive dependence on location accuracy [8]. To ensure that the sample selection covered the research area as much as possible, Zhang et al. established a grid based on the research area and randomly selected training samples for each grid square, which improved the accuracy of grassland classification [9]. However, southwest grassland distribution is mostly distributed on mountains at different altitudes. By only ensuring the spatial uniformity of sample selection it is difficult to fully represent southwest grassland, which may cause serious commission and omission errors. Therefore, reducing this effect by considering the topographic features of grassland distribution in the sample selection process was key for fragmented grassland identification in mountainous regions.

Time series data are used as one of the input features to be able to effectively deal with the problem of "foreign objects with the same spectrum" and have a positive impact on the final classification result [10–18]. Fortunately, there are currently high-temporal-resolution remote sensing data that provide data support for the remote sensing identification of southwest grassland distribution based on a time series. He et al. [19] proved the potential of using Sentinel-1/2 time series data to map the distribution of rice cultivation in cloudy areas. However, it is difficult to establish a complete multispectral time series for image classification under a no-cloud standard in southwest China. Therefore, the establishment of a complete multispectral time series was also key to fragmented grassland identification in cloudy regions.

Additionally, learning the topographic features of grassland distribution in the input features is also critical to improving the accuracy of grassland distribution identification [20]. In many grassland extraction studies, a digital elevation model (DEM) was used as one of the grassland discrimination rules, and then the accuracy of grassland classification was improved [21–23].

However, image classification based on a high number of samples and input features has the limitations of high labor and time costs. The GEE platform can directly call the massive number of images that have been preprocessed and the variety of algorithms that have been packaged, and it has powerful data processing and analysis capabilities that can effectively deal with this problem [24,25]. In recent years, many studies have completed image classification based on time series data and the GEE platform [26–29]. Those studies showed that GEE can provide massive data and cloud computing support for remote sensing research.

Therefore, this study mainly used the GEE platform and considered Zhaotong City, Yunnan Province, as the research area to achieve the accurate identification of grassland

distribution by remote sensing data in a mountainous region. The main innovations include the following: (1) generating the range of sample selection based on five sets of non-homogenous land use/cover data, and then determining samples using the constraint of multiple factors of terrain, secondary land use types and also pure pixels, in order to make sure that the samples can completely cover the various land uses under the premise that the samples are randomly and evenly spatially distributed; (2) a complete multi-spectral time series dataset mainly based on the fusion of Landsat 8 OLI and Sentinel-2A/B was constructed, which was used together with synthetic aperture radar Sentinel-1 time series data and the terrain multi-factor data to enhance the characteristics of ground objects. The above two points improved the separability of ground objects and the accuracy of the classification results.

The structure of this article is as follows: Section 2 introduces the study area and data sources used in the study in detail, including remote sensing data, topographic data, existing related thematic data bases, and verification data. In Section 3, the grassland distribution identification method is introduced, including the sample selection method, input feature selection method, classification method, and verification method. The results of grassland extraction and accuracy assessment are described in Section 4; Section 5 is the discussion around this research; and Section 6 summarizes the conclusions of this article.

## 2. Study Area and Data Sources

### 2.1. Study Area

Southwest China is a vast area that lies south of the Qinling Mountains and the Huai River, and east of the Qinghai–Tibet Plateau. Grassland accounts for 30.51% of the total land area in southwest China. Yunnan Province has the highest proportion of natural grassland area in the region [30].

In this study, Zhaotong City in Yunnan Province (Figure 1) was selected as a demonstration area for the remote sensing identification of grassland distribution. Zhaotong City is located in the hinterland of the Wumeng Mountains in the northeastern part of Yunnan Province, at the junction of the three provinces of Yunnan, Guizhou, and Sichuan. It is located between 102°52′ and 105°19′ E, and 26°55′ and 28°36′ N. It covers an area of 23,000 km$^2$.

The terrain of Zhaotong City is undulating, with high mountains and deep valleys, and there are great differences in climate at different altitudes. The annual average temperature is in the range of 11–21 °C, and the annual average rainfall is in the range of 660–1230 mm. The area covered by forest and grass in the territory accounts for approximately 58% and represents a healthy natural environment [31]. However, the current scale level of animal husbandry in Zhaotong City is relatively low. The proportion of beef and mutton in the city accounts for only 7.34% of the total meat products, while the city has 10,080 km$^2$ of natural pastures, and so has not yet taken advantage of its rich grassland resources [32].

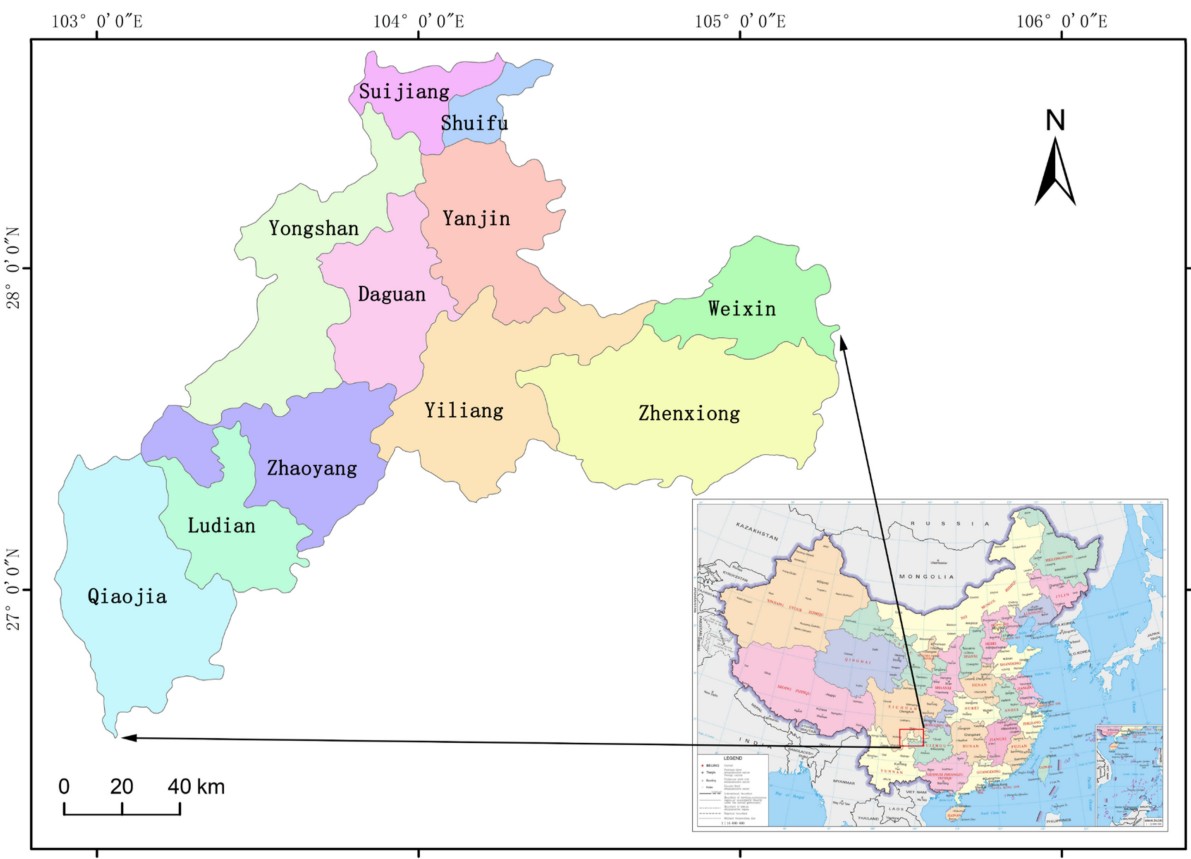

**Figure 1.** Study area of Zhaotong City, Yunnan Province.

*2.2. Data Sources*

The geographic coordinate system used for all data in this study was GCS_WGS_1984.

2.2.1. Remote Sensing Data

Sentinel-1, Landsat 8 OLI, and Sentinel-2A/B, used in this study, were directly archived in the GEE platform. The GF-1 data originated from the China Earth Observation Shared Data Platform. After downloading and processing, they were uploaded to the GEE platform for classification and calculation. Due to the different sensor types of the above data, they are collectively referred to as multi-source remote sensing data.

The calibrated, ortho-corrected Sentinel-1 ground range detected (GRD) scenes were processed using the Sentinel-1 toolbox. Landsat 8 OLI images were level-2 data products after atmospheric correction by the Landsat Surface Reflectance Code (LaSRC) method. Sentinel-2A/B images were 2A-level data products after atmospheric correction by the Sentinel-2 Level-2A atmospheric correction processor (Sen2cor) method. The geometric correction of GF-1 data was accomplished with Landsat's simultaneous precision correction data.

Specifically, in order to reduce the impact of outliers on the classification results, Sentinel-1 is a mosaic of the median value of Sentinel-1 images month by month, which is composed of 12 images. Sentinel-2A/B and Landsat 8 OLI images were fused on the GEE platform. First, Sentinel-2A/B and Landsat 8 OLI images, after masking cloud cover in 2019, 2020, and 2021, were fused monthly at the median value, and then the median value of the corresponding month of the 2019 and 2021 fusion images were used to supplement the 2020 fusion images, namely, the 2020 fusion images were supplemented with the fusion images of the adjacent year according to the corresponding month. However, limited by the number of images and cloud cover, only the fusion images of February, March, April, June, August, and November 2020 covered almost the entire study area, with a coverage

of 99.91%, 99.99%, 100%, 99.92%, 100%, and 99.97%, respectively. The GF-1 image on 27 August 2020 was used. Additional information on the above data is shown in Table 1.

**Table 1.** Description of the remote sensing data used in this study.

| Sensor Used | Bands | Descriptions | Resolution | Input Features |
|---|---|---|---|---|
| Sentinel-1 | VV | 5.405 GHz | 10 m | Radar time series data |
| | VH | 5.405 GHz | 10 m | |
| Landsat 8 OLI | Blue | 452–512 nm | 30 m | |
| | Green | 533–590 nm | 30 m | |
| | Red | 636–673 nm | 30 m | |
| | NIR | 851–879 nm | 30 m | |
| | SWIR1 | 1566–1651 nm | 30 m | |
| | SWIR2 | 2107–2294 nm | 30 m | |
| Sentinel-2A/B | Blue | 496.6 nm (S2A)/492.1 nm (S2B) | 10 m | Multispectral time series data |
| | Green | 560 nm (S2A)/559 nm (S2B) | 10 m | |
| | Red | 664.5 nm (S2A)/665 nm (S2B) | 10 m | |
| | NIR | 835.1 nm (S2A)/833 nm (S2B) | 10 m | |
| | SWIR1 | 1613.7 nm (S2A)/1610.4 nm (S2B) | 20 m | |
| | SWIR2 | 2202.4 nm (S2A)/2185.7 nm (S2B) | 20 m | |
| GF-1 | Blue | 450–520 nm | 16 m | |
| | Green | 520–590 nm | 16 m | |
| | Red | 630–690 nm | 16 m | |
| | NIR | 770–890 nm | 16 m | |

Considering the connectivity between multi-source data, the nearest neighbor interpolation method was used to resample all bands with non-10 m resolution to 10 m.

### 2.2.2. Terrain Data

A DEM of Zhaotong City, with a spatial resolution of 15 m, derived originally from the Synthetic Aperture Radar Satellite in 2000, was obtained as commercial data by this study, and was used to calculate the slope and aspect data using the ArcGIS 10.5 software. The three datasets constituted the terrain multi-factor data required for this study, resampling to 10 m. The three terrain datasets were used as input features of the classification model on the GEE platform.

### 2.2.3. Use of Existing Thematic Databases

Six non-homogenous remote sensing monitoring data that reflected grassland information were used to determine the sample selection range. The spatial resolution and mapping accuracy of each data product are shown in Table 2.

**Table 2.** Details of the six non-homogenous remote sensing monitoring datasets used in this study.

| Name | Resolution | Mapping Accuracy |
|---|---|---|
| 1:100,000 land use data [33] | 30 m | 85% |
| GlobeLand30 data [34] | 30 m | 83.50% |
| CGLOPS-1 data [35] | 100 m | 80% |
| GLC_FCS30 data [36] | 30 m | 82.50% |
| FROMLC data [37] | 10 m | 72.76% |
| China 1:1,000,000 vegetation map [38] | — | 64.8% |

With reference to the classification system of the GlobeLand30 data, combined with the actual land use situation in the study area, the classification system used in this study was cultivated land (10), forest (20), grassland (30), water bodies (60), impervious surfaces (80), and bare land (90). The classification system of other data products (Table 2) was transformed to this classification system for sample selection.

### 2.2.4. Verification Data

This study collected grassland verification samples through field surveys, and non-grassland verification samples and supplementary grassland verification samples were determined through an expert visual interpretation of high-spatial resolution images on the GEE platform. There were 112 non-grassland verification samples and 117 grassland verification samples. The spatial distribution of the verification samples is shown in Figure 2.

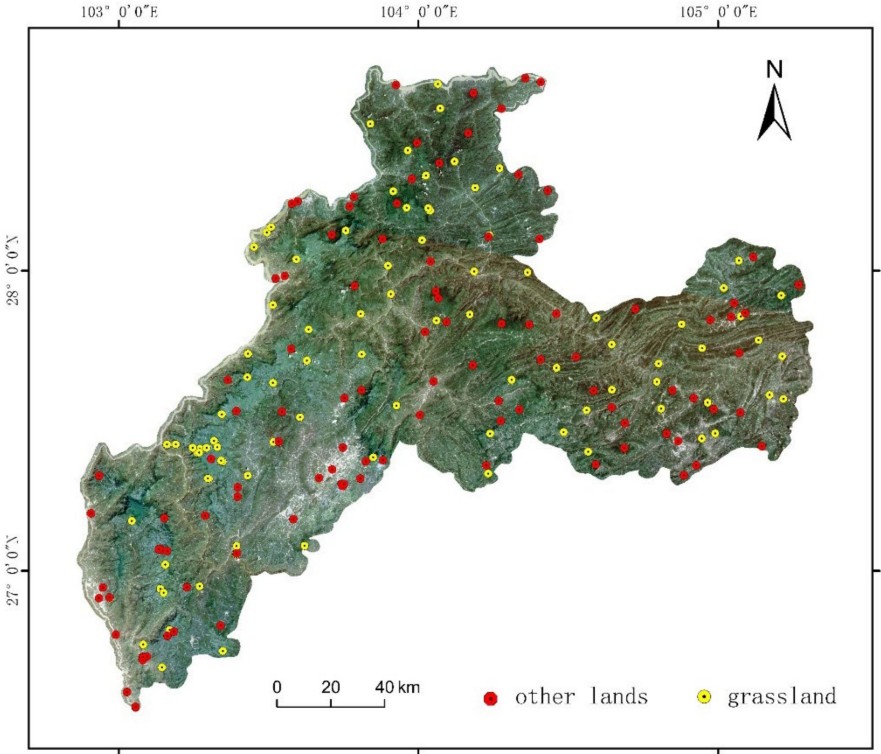

**Figure 2.** Location of verification samples.

### 3. Methods

The overall workflow used for grassland distribution identification in Zhaotong City in 2020 is presented in Figure 3, consisting of the following steps: (1) samples; (2) input features; (3) experimental design; and (4) classification and validation.

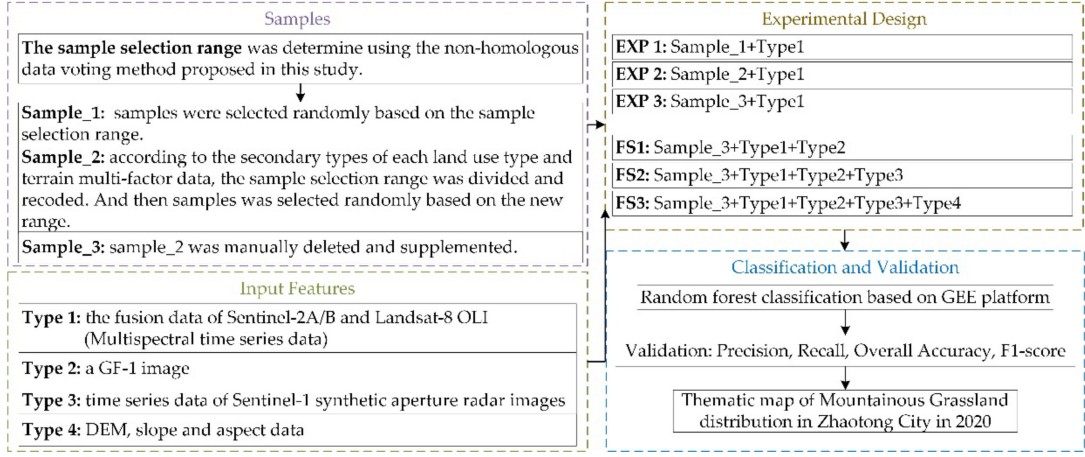

**Figure 3.** Flow chart of the remote sensing grassland identification method.

### 3.1. Sample Selection

The quantity and quality of sample selection have a greater impact on the classification results than the classification algorithm does [39]. In order to accurately identify the distribution of grassland in southwest China, this study utilized the following process of sample selection:

(1)  Determine the sample range using a non-homologous data-voting method:

There are many mountains and hills in southwest China, which makes it difficult to collect samples in the field and leads to high time and labor costs. Therefore, the non-homologous data-voting method was proposed in this study to determine the sample range. Specifically, 1:100,000 land use data, GlobeLand30 data, CGLOPS-1 data, GLC_FCS30 data, and FROMLC data were collected, and then pixels with the same classification of cultivated land, forest, grassland, impervious surfaces, water bodies, and bare land in the above data were used as the sample selection range. The sample was randomly generated within the sample selection range, which is called sample_1 in this study.

(2)  Divide and recode the sample selection range according to the secondary land use types and terrain multi-factor data:

Referring to the 1:100,000 land use data, the sample selection range of cultivated land, grassland, and impervious surfaces was divided according to the corresponding secondary land use types. Referring to the CGLOPS-1 data, the sample selection range of forest was divided according to the corresponding secondary land use types. Additionally, the sample selection range of bare land, water bodies and impervious surfaces was not divided according to the secondary land use types. The specific types are shown in Table 3.

**Table 3.** Discrete classification types for dividing the sample selection range.

| Primary Land Use Types | Secondary Land Use Types |
| --- | --- |
| Cultivated land | Mountain paddy; hilly paddy; plain paddy; paddy with slopes above 25°; mountain dryland; hilly dryland; plain dryland; dryland with slopes above 25° |
| Grassland | High coverage grassland; medium coverage grassland; low coverage grassland |
| Impervious surfaces | Urban land; rural residential land; industrial and construction land |
| Forest | Closed forest, evergreen needle leaf; closed forest, deciduous needle leaf; closed forest, evergreen, broad leaf; closed forest, deciduous, broad leaf; closed forest, mixed; closed forest, unknown; open forest, evergreen needle leaf; open forest, deciduous needle leaf; open forest, evergreen broad leaf; open forest, deciduous broad leaf; open forest, mixed; open forest, unknown |

The DEM was reclassified into 0–1400 m (low land), 1400–2100 m (medium land), and 2100 m and above (high land), and the slope data were reclassified into 0–15° (gentle slope), 15–25° (medium slope), and 25° and above (steep slope). The aspect data were reclassified into two classes: sunny slope and shady slope. Only the sample selection range of cultivated land, grassland, and forest was divided according to the multi-factor terrain data.

Finally, the sample selection range was divided and recoded according to the above data, except for water bodies and bare land. Cultivated land, forest, grassland, and impervious surfaces are divided into categories of 82, 120, 51, and 3. As shown in Figure 4, the type code after the division was composed of the secondary land use type and the corresponding category codes of the DEM, slope, and aspect after reclassification.

The purpose of dividing and recoding the sample selection range is to make the sample selection results more complete and representative of the study area. Additionally, taking into account the practical application of determining the range of grassland distribution, this code was only used for sample selection, and the actual classification type was consistent with the primary land use type of the sample selection.

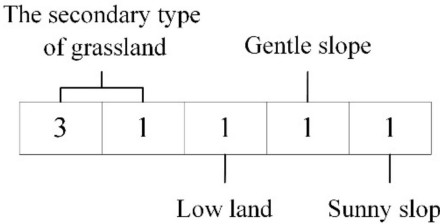

**Figure 4.** An example of how the sample selection range was recoded.

(3)    Determine the final sample selection range by filtering pure pixels and then generate random samples:

To avoid the problem of excessive dependence on the location accuracy of mixed pixels and heterogeneous grass samples, the algorithm, designed in this study, used a 900 m$^2$ square as the standard, and then traversed the type-encoded raster data (the sample selection range generated by (2)) sequentially. Restricted by the area threshold, after filtering pure pixels, the number of sub-categories of cultivated land, forest, grassland, and impervious surfaces are 66, 116, 46, and 3, respectively. As shown in Figure 5, sample squares with an area of 900 m$^2$, different types of codes, and non-overlapping areas were selected as the final sample selection range. In addition, considering the randomness of sample selection, the ArcGIS 10.5 software was used to create random sample points for each type of code by controlling the minimum allowable distance between sample points of the same type of code according to the final sample selection range. On the basis of the number of samples in each primary class, the number of samples in each subclass was proportional to the number of corresponding pure pixels. Additionally, they were then merged according to the primary type code, resulting in six types of random samples, which are referred to as sample_2 in this study.

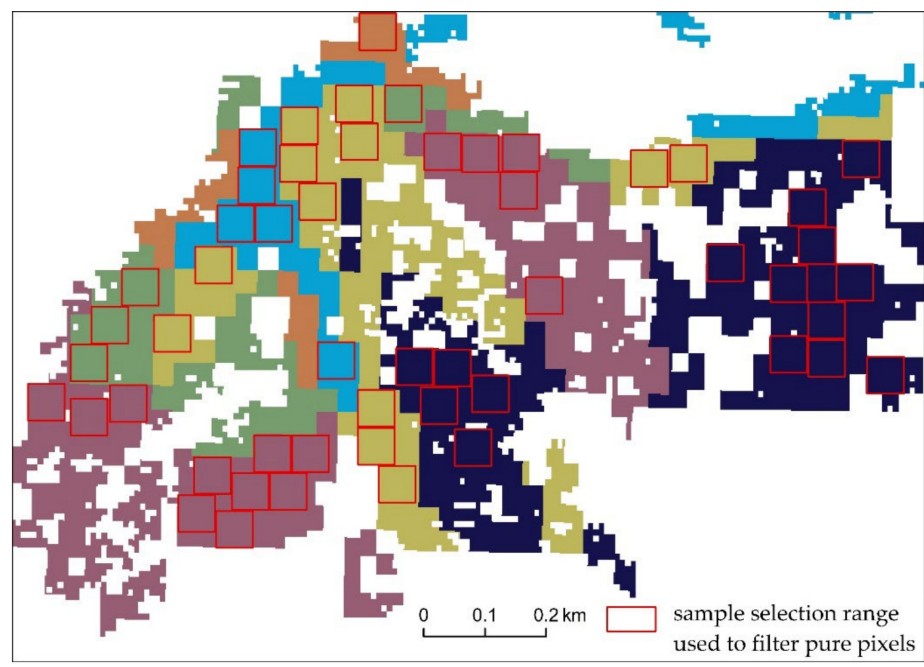

**Figure 5.** The example of the final sample selection range used to filter pure pixels and to obtain random samples.

(4)    Delete and supplement samples manually on the basis of (3):

With the help of high-spatial-resolution images, a normalized difference vegetation index (NDVI) time series curve, and China's 1:1 million vegetation map on the GEE platform, partial samples of (3) were supplemented and deleted. Due to the non-homologous

data-voting method, samples were missing in areas with inconsistent classification results of non-homologous data and, therefore, were supplemented. Additionally, partial samples of vegetation types were deleted with reference to China's 1:1 million vegetation map in order to improve the reliability of the vegetation samples. The sample generated after manual intervention is called sample_3.

In addition, in order to reflect the advantages of the sample selection method proposed in this study, multi-spectral time series based on the fusion of Landsat 8 OLI and Sentinel-2A/B were used as input features to test the influence of sample_1, sample_2, and sample_3 as samples on the classification results, respectively. The number of samples of each type was proportional to the area of that type in Zhaotong, and the number of samples for the six classification types in the above three samples is the same.

### 3.2. Input Feature Selection

To compare the performances of different features, we designed three feature scenarios as shown in Table 4: (1) FS1 was designed to examine the ability of multispectral time series data (multispectral fusion images and GF-1) to identify southwest grassland distribution; (2) FS2 was designed to explore whether integrated multispectral and radar time series data (Sentinel-1) can improve the classification accuracy; and (3) FS3 was designed to explore the ability of the terrain multi-factor data (DEM, slope, and aspect) in improving the separability between other lands and grassland. Additionally, in the process of conducting the above three feature scenarios, sample_3 was used as the training samples of the classification model.

**Table 4.** Scenarios design based on input feature selection.

|  | FS1 | FS2 | FS3 |
|---|---|---|---|
| Multispectral time series data | √ | √ | √ |
| Radar time series data |  | √ | √ |
| The terrain multi-factor data |  |  | √ |

### 3.3. Random Forest Classification on the GEE Platform

As a current mainstream machine learning model, the random forest model can predict the effects of thousands of explanatory variables. It uses decision trees as a unit and aggregates multiple decision trees for classification, which can effectively solve the classification problem of a large amount of high-dimensional data [40]. Therefore, we used the random forest model to classify the input features.

Due to the large number of input features and samples, these processes were all completed by the high-performance cloud computing GEE platform. There is a need to adjust the number of decision trees when we invoke the random forest algorithm of the GEE platform. Considering the computational efficiency, we selected the number of decision trees with relatively high overall accuracy from the number of 1–200 decision trees as the optimal parameter. Other parameters in the classifier were set as the default. After the ablation test, the optimal parameter of the model was identified, the classification model was determined, and the grassland distribution range of Zhaotong City was obtained in each experiment.

### 3.4. Verification of Grassland Extraction Results

To evaluate the accuracy of remote sensing in identifying grassland distribution, we combined the six types of classification systems into two: grassland and other lands. Additionally, the four evaluation indicators of precision, recall, overall accuracy, and F1 score were calculated through the verification data (Section 2.2.4) to assess the grassland extraction results.

Precision refers to the proportion of extracted positive examples (grassland) in all extraction results, and recall refers to the proportion of extracted positive examples in all positive examples. Overall accuracy refers to the proportion of all correctly extracted

examples from all examples. F1 score refers to the harmonic average of precision and recall. The closer the values of the above four indicators are to 1, the better the results of the grassland distribution recognition. The specific formula are as follows:

$$\text{Precision} = \frac{TP}{FP + TP} \tag{1}$$

$$\text{Recall} = \frac{TP}{FN + TP} \tag{2}$$

$$\text{Overall Accuracy} = \frac{TP + TN}{FN + TP + FP + TN} \tag{3}$$

$$\text{F1 score} = \left(1 + \beta^2\right)\frac{\text{Precision} \times \text{Recall}}{\text{Precision} + \text{Recall}}, \ \beta = 1 \tag{4}$$

Here, TP, FP, FN, and TN are shown in Table 5.

**Table 5.** Confusion matrix.

| Predicted Value \ Actual Value | Grassland | Other Lands |
|---|---|---|
| Grassland | True Positive (TP) | False Positive (FP) |
| Other lands | False Negative (FN) | True Negative (TN) |

## 4. Results and Analysis

### 4.1. Results and Analysis of Sample Selection

The sample selection in this study satisfied the requirements of randomness, uniformity, and completeness, and could completely represent various realistic manifestations of different land use types under different terrain conditions. The final sample selection is shown in Figure 6 and the sample numbers of cultivated land (10), forest (20), grassland (30), water bodies (60), impervious surfaces (80), and bare land (90) were 1162, 627, 500, 65, 134, and 39, respectively.

By comparing the classification results of EXP1, EXP2, and EXP3 with the high-resolution images of Google Earth, it was found that the grassland omission problem of EXP1 was mainly distributed in the mountains in the north and east of Zhaotong City, particularly on the steep slopes of the mountains. This is due to the fact that sample_1 was randomly generated within the sample selection range, without division, according to the terrain multi-factor data. Hence, it does not have a complete representation of grassland samples and is also unevenly distributed in the study area (as shown in Figure 7).

As shown in Figure 8, the classification results of EXP1 had the problem of omitting grassland into cultivated land in the steep slopes of the mountains, which was improved in EXP2 and EXP3. Compared with EXP1, the omission errors of grassland in EXP2 and EXP3 were reduced by 0.1759 and 0.2308, respectively, indicating that it was both necessary and effective for sample selection to follow the principle of integrity. Although EXP2 also obviously improved the grassland omission problem, it still had the problem of omitting grassland into cultivated land, and EXP3 identified grassland distribution more accurately than EXP2. This is because the samples of EXP3 were more complete and more evenly distributed in space after manual intervention in sample selection. Among them, the overall accuracy of EXP3 was the highest, which was 0.1223 higher than that of EXP1 and 0.0393 higher than that of EXP2 (as shown in Table 6).

In summary, the classification results and accuracy based on sample selection show that representative and spatially evenly distributed samples can improve the separability of cultivated land and grassland to a great extent.

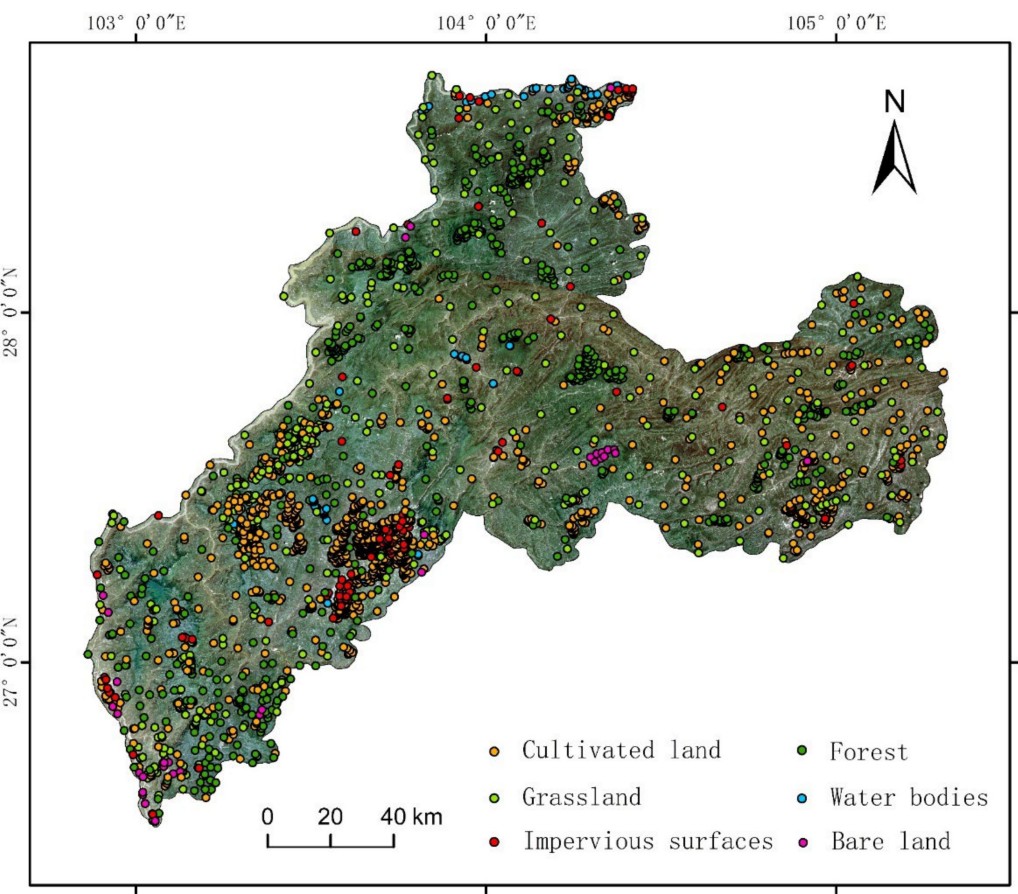

**Figure 6.** Sample selection results.

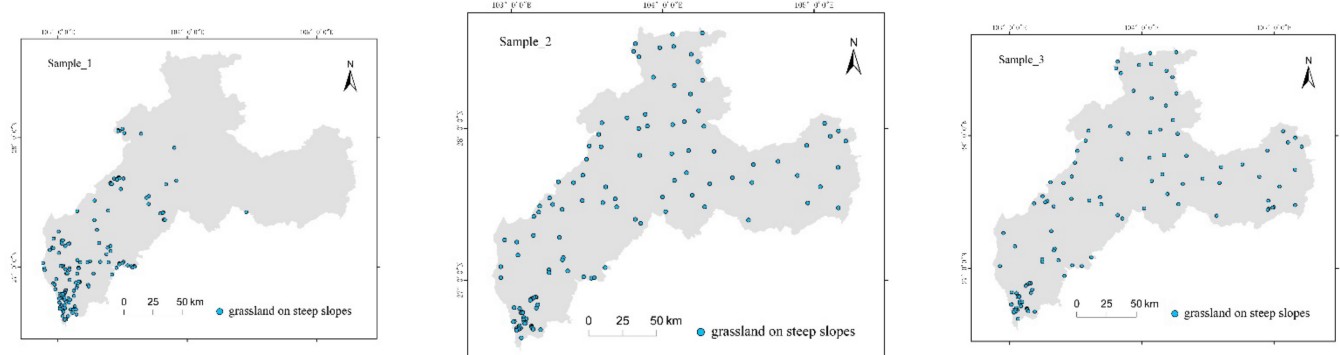

**Figure 7.** Sample selection for grassland on steep slopes in sample_1, sample_2, and sample_3.

**Table 6.** Classification accuracy for the grassland distribution identification based on sample selection.

|       | Precision | Recall | Overall Accuracy | F1 Score |
|-------|-----------|--------|------------------|----------|
| EXP1  | 0.9375    | 0.3846 | 0.6725           | 0.5455   |
| EXP2  | 0.9296    | 0.5641 | 0.7555           | 0.7021   |
| EXP3  | 0.9730    | 0.6154 | 0.7948           | 0.7539   |

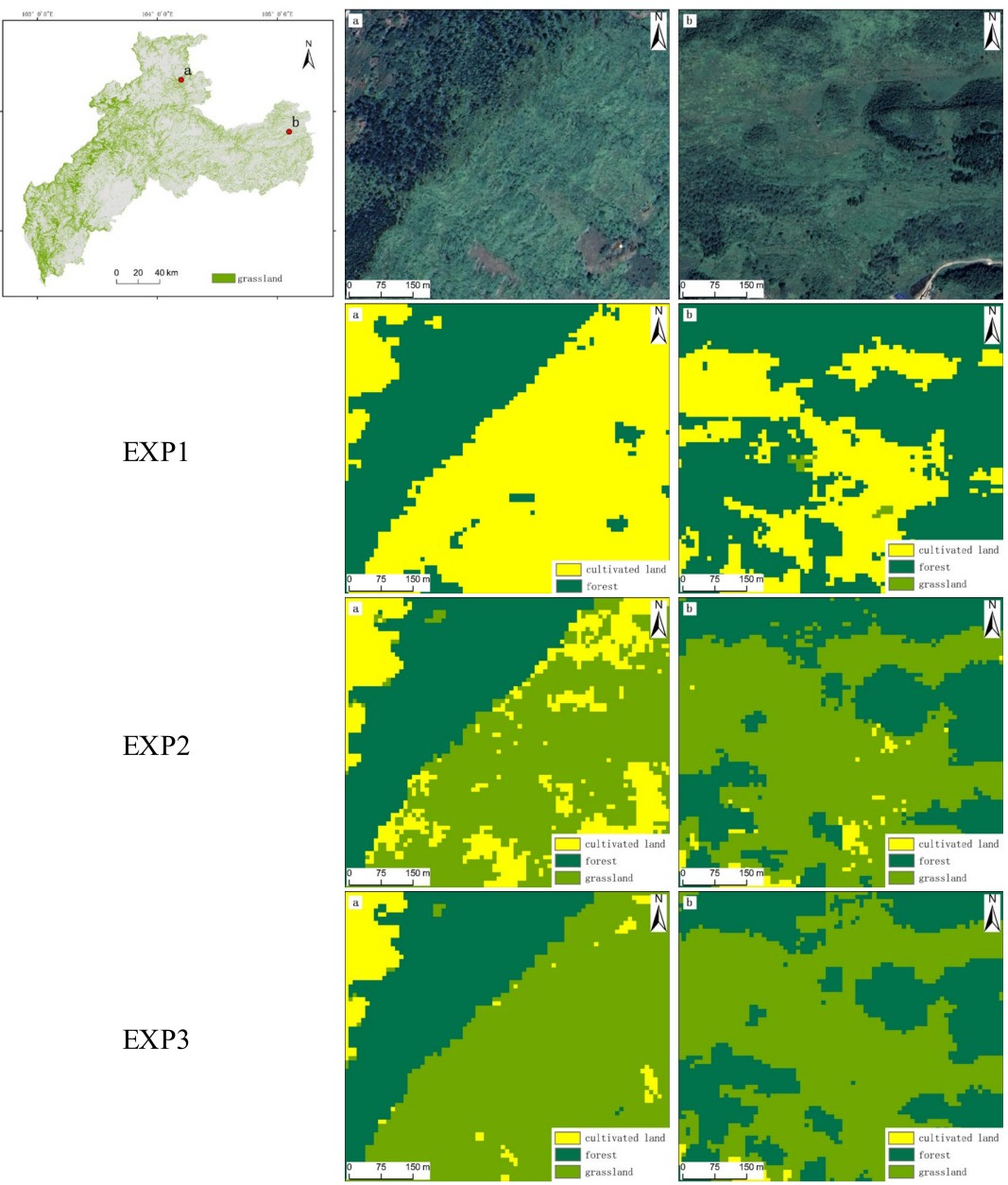

**Figure 8.** Examples of EXP1, EXP2, and EXP3 classification results. Region a is located on sunny slopes in the low land with steep slopes, and region b is located on sunny slopes in the middle land with steep slopes.

### 4.2. Results and Analysis of Input Feature Selection

By comparing the classification results of FS1, FS2, and FS3 with the high-resolution images of Google Earth, it was found that the classification results of FS1 had the problem of omitting grassland into cultivated land in the low terrain areas of Zhaotong City. This problem was improved after adding radar time series data (FS2) and the terrain multi-factor data (FS3) as input features. FS3 showed that adding radar time series and the terrain multi-factor data together had greater separability between grassland and cultivated land than using sentinel-1 only (FS2). In addition, FS2 showed that there was a problem of omitting a small amount of grassland into forest on steep slopes, while FS3 had better performance in distinguishing forest and grassland on steep slopes. After adding the terrain multi-factor data as part of input features (FS3), the omission errors of grassland were reduced by 0.11 more than those of FS2. Examples of FS1, FS2, and FS3 classification results are shown in Figure 9.

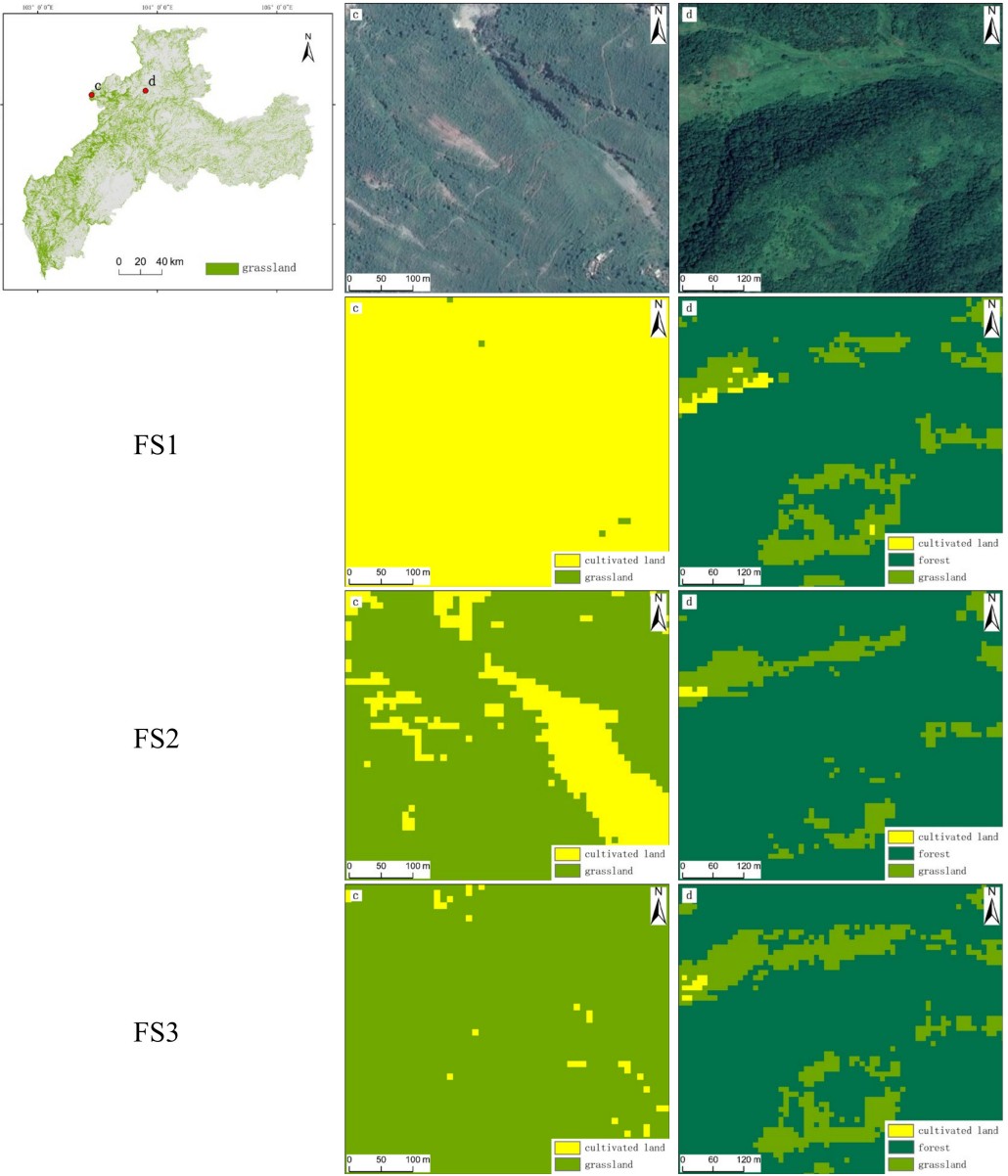

**Figure 9.** Examples of FS1, FS2, and FS3 classification results. Region c is located on sunny slopes in the low land with steep slopes, and region d is located on shady slopes in the middle land with steep slopes.

The final result of input feature selection is FS3. Classification results and accuracy based on input feature selection showed that learning the features of ground objects as comprehensively as possible is beneficial to the accurate identification of ground object distribution. The combined use of multispectral and radar time series is helpful in distinguishing grassland and cultivated land in the lowland with steep slopes. The specific classification accuracy is shown in Table 7.

**Table 7.** Classification accuracy for the grassland distribution identification based on input feature selection.

|  | Precision | Recall | Overall Accuracy | F1 Score |
|---|---|---|---|---|
| FS1 | 0.9398 | 0.6667 | 0.8079 | 0.7800 |
| FS2 | 0.9873 | 0.6667 | 0.8253 | 0.7959 |
| FS3 | 0.9891 | 0.7778 | 0.8821 | 0.8708 |

### 4.3. Results and Analysis for the Grassland Distribution Identification

The thematic map generated in this study of the grassland distribution (Figure 10) in Zhaotong City in 2020, with an overall accuracy of 88.21%, has been used to provide reliable grassland resource background data for the development of local sustainable grassland husbandry. The difficulty of accurately identifying the grassland distribution in southwest China due to the complex terrain and cloudy and rainy climate has been effectively addressed.

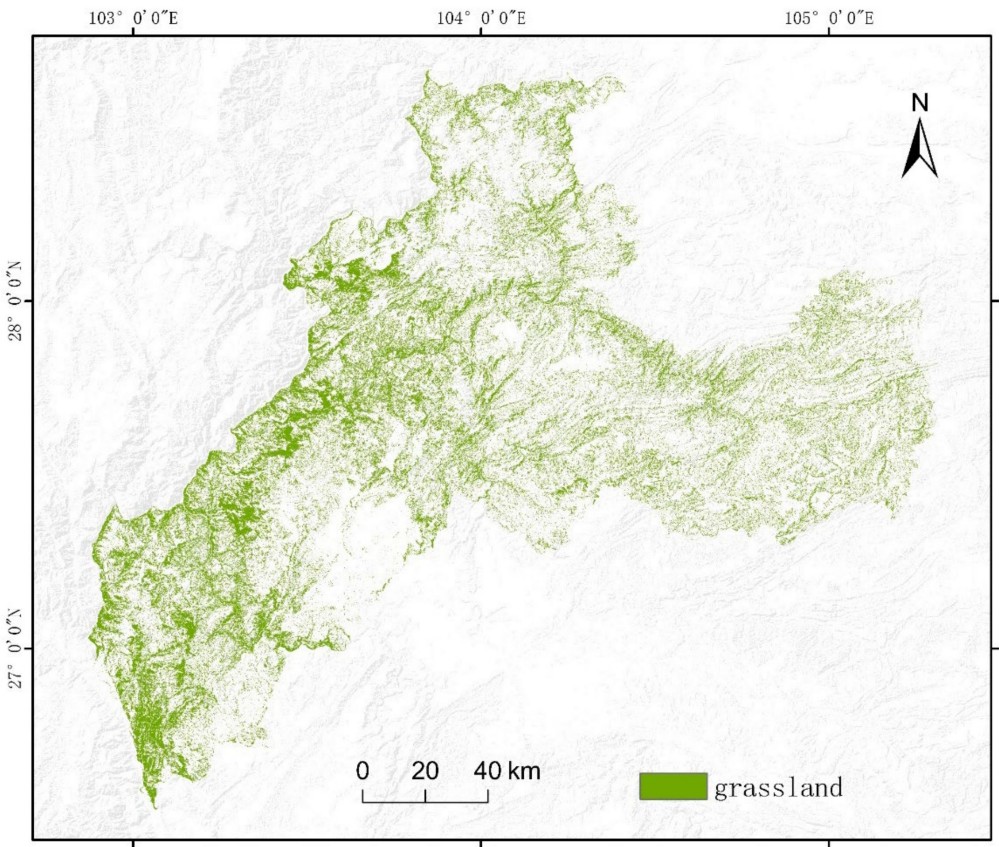

**Figure 10.** Grassland distribution remote sensing identification map of Zhaotong City, Yunnan Province in 2020.

In the grassland extraction results, the smallest patch area is approximately 88 m$^2$, and the largest patch area is 22,819,650 m$^2$, reflecting that the remote sensing identification method of mountainous grassland distribution in this paper has advantages in identifying broken grassland patches, and is suitable for application in mountainous and hilly areas.

### 5. Discussion

The fragmented distribution of grassland in southwest China is an important reason for its difficulty in accurate identification. Additionally, remote sensing identification is also limited by the lack of no-cloud images. Therefore, this study attempted to deal with these problems in two aspects: sample selection and input feature selection.

In terms of sample selection, the reliable, rapid, and reproducible collection of training samples is a challenge for land cover classification [41], especially in a fragmented mountainous region. The non-homologous data-voting method promoted in this paper reduces the errors and uncertainties in the training samples extracted from a single source dataset. Since the sample selection range, determined by the non-homologous data-voting method, only accounts for 19% of the study area, only 29 grassland verification samples and 56 non-grassland verification samples are included in this range. Among them, two grassland verification samples were wrongly classified as non-grassland, and two non-grassland

validation samples were wrongly classified as grassland. Additionally, from the results and accuracy of grassland identification, it is reliable and available to determine the training samples based on the non-homologous data-voting method. Compared with the method of collecting samples in the field, this method not only made the samples fast and easy to obtain, but also made samples objective and not subject to human influence. However, a limitation is that the non-homologous data-voting method limits the sample selection range, resulting in the lack of samples in some areas, which then need to be supplemented manually. In follow-up work, we can use other data products with higher accuracy to complete sample selection without manual intervention.

In terms of input feature selection, the construction of multispectral time series data used the median data of the corresponding months in the two years before and after to supplement the 2020 data, which not only improved the coverage (only 0.21% of the study area has no data), but also maintained the geographic rationality of the time series spectrum characteristics of ground objects. Additionally, multispectral time series data can make up for the negative impact of hill shadows in radar data on grassland classification, to a certain extent [42,43]. The grassland extraction results show that the grassland distribution has significant terrain characteristics: in Zhaotong, approximately 64% of the grassland is distributed in the area with an altitude of more than 1400 m, approximately 75% of the grassland is distributed in the area with a slope of more than 15°, and approximately 50% of the grassland is distributed on shady slopes. Therefore, taking the terrain multi-factor data as one of the input features is conducive to fully learning the terrain characteristics of ground objects, thereby reducing the commission and omission errors of grassland.

In addition, the DEM data with a resolution of 15 m, used in sample selection and input feature selection, can be replaced with ALOS PALSAR products with a resolution of 12.5 m (URL: https://earth.esa.int/eogateway/catalog/alos-palsar-products?text=ALOS+PALSAR+RTC, accessed on 28 February 2022) when applying the remote sensing method to accurately identify grassland distribution; this may make the identification of grassland distribution more accurate.

## 6. Conclusions

The remote sensing data product of Zhaotong City in 2020 solved the problem of grassland borders being indistinguishable in the identification of grassland distribution, and therefore was able to meet the needs of prefecture-level cities. It will enable the authorities in Zhaotong City to understand the distribution of their grassland resources to accelerate the development of sustainable grassland husbandry. Using the GEE platform, which has strong computing power, a large amount of global-scale satellite data, and the capability for online visual calculation and analysis, the methods in this study were suitable for monitoring dynamic changes in grassland and for assessing the degradation of grassland over several decades.

In summary, compared with previous remote sensing grassland monitoring data, the remote sensing grassland distribution identification data product produced in this study can better meet the needs of Zhaotong City for the development of sustainable grassland husbandry. The specific conclusions are as follows.

(1) Sample selection should follow the principles of completeness and randomness. If the sample selection range is not divided according to the secondary land use types and terrain multi-factor data, the results of random sample selection in the study area will not be fully representative. This can cause poorer classification results in areas lacking samples that conform to the realistic representation of primary land use types. In this study, complete sample selection mainly reduced the omission errors of grassland and effectively solved the problem of the grassland distribution being difficult to accurately identify due to the complex topography of southwest China.

(2) The combined use of all available multispectral and radar data has the potential to identify the grassland distribution in mountainous fragmented terrain, and terrain characteristics are vital to mountainous grassland identification. This study used

multispectral and radar time series data as input features, which effectively solved the problem of the grassland distribution being difficult to accurately extract due to cloud cover and heavy rain in southwest China. The input features applied in this study enabled the model to learn the time spectrum characteristics of radar and optical images, and the topographic features of southwest grassland, which improved the separability of ground objects.

(3) The random forest model is suitable for dealing with the classification problem of multiple input features, which can be efficiently calculated and classified by the GEE cloud computing platform. In this study, there were 2527 sample points (including training and test samples) and 67 bands of input features. Experiments have shown that the random forest model can effectively learn multiple input features, and that the GEE platform only takes approximately 2–3 min to identify the optimal parameters (number of decision trees) for the model. Therefore, with a small time cost, a remote sensing thematic map of the grassland distribution in Zhaotong City in 2020 was obtained using the GEE platform.

**Author Contributions:** All authors contributed in a substantial way to the manuscript. Y.Y. and Q.W. conceived of, designed, and performed the research and wrote the manuscript. X.Z., S.L., B.H. and K.Z. contributed to verification data collection through field surveys and manuscript revision. All authors have read and agreed to the published version of the manuscript.

**Funding:** This research was funded by the Strategic Priority Research Program of Chinese Academy of Sciences (Grant No. XDA26050301-01) and the Inner Mongolia Autonomous Region Science and Technology Achievement Transformation Special Project (Grant No. 2020CG0123).

**Data Availability Statement:** If you need the thematic map of grassland distribution in Zhaotong City in 2020 generated by this study, please contact email: yuanyixin19@mails.ucas.ac.cn.

**Acknowledgments:** The authors would like to thank the Google Earth Engine platform for providing us with a free computing platform and free data and we also thank anonymous reviewers for their insightful advice.

**Conflicts of Interest:** The authors declare no conflict of interest.

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
