# Peer review of "Identifying Grassland Distribution in a Mountainous Region in Southwest China Using Multi-Source Remote Sensing Images"

_remotesensing, doi:10.3390/rs14061472_

Round 1
Reviewer 1 Report
The paper is devoted to an important task of grassland identification for mountanous region using different types of remore sensing data. The authors clearly describe the task which is really challenging and worth solving. Meanwhile, the paper suffers from several drawbacks, namely:
1) The goal is not clearly formulated;
2) The novelty is not clearly described, the paper structure is not given in Introduction:
3) There is an unclear sentence in Lines 48-50;
4) Please explain the term "multi-source". I guess the meaning but...
5) Line 147 - median value? Why? Please explain here;
6) Line 163 - who obtained the DEM? Was it available? Or the authors have obtained it somehow?
7) SAR data in mountanous regions can produce shadowed zones. How have the authors taken this into account?
8) How to explain the differences in performance cgaracteristics of EXP2 and EXP3 in Table 2?
Author Response
Thanks teacher for your advice and guidance, please see the attachment for specific responses.

Reviewer 2 Report
The paper deals with the use of multi-source data for determining the distribution of grassland in Southwest China. The subject matter has a basis of interest but some aspects of the paper need to be thoroughly revised. The first aspect that needs to be revised is a more correct use of the English language. The construction of sentences, the organisation of discourse and the inappropriate use of certain terms make reading very tiring. From the methodological point of view, the paper underlines the advantages of the methodology adopted for the selection of the samples used to train the machine learning method employed. However, this method is not explained clearly and exhaustively. There is also a lack of information and considerations necessary to robustly compare the results obtained in the different experiments. For example, it is not clear how many samples per class are considered for each of the proposed sampling methods (this is important to evaluate if the classification has been done with unbalanced classes and if there are differences in the imbalance between the different datasets); whether the samples of the second method are a subset of the first, or a different one? What is the purpose of using subclasses for the choice of samples when you then state that subclasses were not considered for classification?; what is the precise relationship between the numbering of the steps of the sample selection method in the descriptive part (1,2,3,4) and those proposed in the comparison (1, 2, 3).
With regard to the machine learning method used, random forest, no details are provided either on the hyperparameters used (it is assumed that the default ones established by the GEE platform) or the criterion with which the only non-default hyperparameter was optimised. A further aspect that is not clear is the type of classification carried out: Was the training dataset used for a binary classification (grassland, non-grassland), or a multiclass type (as it seems from some figures) and then the classes were aggregated for performance evaluation on the validation dataset?
Have the authors considered the effects of imbalance (different number of samples) between classes in the training phase on the classification performance of the validation dataset, which is almost balanced? Does the balance of classes in the various training datasets conform to the actual prevalence of the various classes in the territory considered for the realisation of the final map? Are the training and validation datasets disjointed?
The authors also state that the joint use of the five types of classification used for the choice of samples (the ones shown in Table 2) allows an error rate of 2.36% (I have doubts about this evaluation, both from a methodological and numerical point of view). The authors should verify the accuracy of this statement by checking how the points in the validation dataset would be classified by this way. This would allow a robust assessment of the advantages of the proposed methodology over simply using the "joint" classification with the sources shown in Table 2.
Author Response

(The authors gave the same response as above.)

Round 2
Reviewer 1 Report
I am satidfied by the corrections done.
Author Response
Thank you teacher for your affirmation and help!
Reviewer 2 Report
The authors addressed all the issues highlighted to the previous version. The authors have responded sufficiently and comprehensively to the proposed questions and have amended the text of the paper consistently. As far as the English language is concerned, it should be noted that the second version of the manuscript contains numerous corrections (additions) but does not show the deleted parts. This makes it very difficult to assess this aspect. It should also be noted that for many sentences the correction appears more incorrect than in the original version. This is probably due to an incorrect version of the attached file.
Author Response
Thanks teacher for your reminder and advice. I have carefully checked and compared the revised manuscript after the Major revised with the manuscript after the English editing service, and made revisions to the omissions according to the English editing service's opinion.